# Genetic Dissection of Apomixis in Dandelions Identifies a Dominant Parthenogenesis Locus and Highlights the Complexity of Autonomous Endosperm Formation

**DOI:** 10.3390/genes11090961

**Published:** 2020-08-20

**Authors:** Peter J. Van Dijk, Rik Op den Camp, Stephen E. Schauer

**Affiliations:** 1Keygene N.V., Agro Business Park 90, 6708 PW Wageningen, The Netherlands; rik.op-den-camp@keygene.com; 2Keygene Inc., Rockville, MD 20850, USA; stephen.schauer@keygene.com

**Keywords:** apomixis, diplospory, parthenogenesis, autonomous endosperm, genetics, *Taraxacum*, dandelion

## Abstract

Apomixis in the common dandelion (*Taraxacum officinale*) consists of three developmental components: diplospory (apomeiosis), parthenogenesis, and autonomous endosperm development. The genetic basis of diplospory, which is inherited as a single dominant factor, has been previously elucidated. To uncover the genetic basis of the remaining components, a cross between a diploid sexual seed parent and a triploid apomictic pollen donor was made. The resulting 95 triploid progeny plants were genotyped with co-dominant simple-sequence repeat (SSR) markers and phenotyped for apomixis as a whole and for the individual apomixis components using Nomarski Differential Interference Contrast (DIC) microscopy of cleared ovules and seed flow cytometry. From this, a new SSR marker allele was discovered that was closely linked to parthenogenesis and unlinked to diplospory. The segregation of apomixis as a whole does not differ significantly from a three-locus model, with diplospory and parthenogenesis segregating as unlinked dominant loci. Autonomous endosperm is regularly present without parthenogenesis, suggesting that the parthenogenesis locus does not also control endosperm formation. However, the high recovery of autonomous endosperm is inconsistent with this phenotype segregating as the third dominant locus. These results highlight the genetic complexity underlying apomixis in the dandelion and underline the challenge of introducing autonomous apomixis into sexual crops.

## 1. Introduction

Apomixis is a form of reproduction in the flowering plants in which the seeds are clones of the mother plant [1,2]. Apomixis, if introduced into the hybrids of otherwise sexually reproducing crops, will revolutionize plant breeding and agriculture because apomixis allows the one-step fixation of any valuable trait (e.g., heterosis or hybrid vigor), irrespective of the genetic complexity of the trait, for all subsequent generations [3,4,5,6]. This will reduce the time and cost of varietal development and is necessary for finding solutions to the immense and acute problems of population growth, changing climates, and the biodiversity crisis. Apomixis, however, does not occur in major crops and is rare in wild plant species. One promising way to introduce apomixis into crops is to reverse-engineer wild apomictic species: to genetically dissect natural apomixis, to clone natural apomixis genes, to identify the sexual orthologs of apomixis genes, and to modify these orthologs to apomictic versions in the crops of interest.

In the case of gametophytic apomixis, which may be the most straightforward approach to introducing apomixis into crops, a diploid egg cell is formed that is genetically identical to the mother plant and then transitions into an embryo without fertilization. Gametophytic apomixis is a rare reproductive system but widely distributed phylogenetically. One of the best-known apomicts is the common dandelion, *Taraxacum officinale* in the *Asteraceae* family, in which sexual diploid (2*n* = 2*x* = 16) and apomictic polyploid (mainly triploid, 2*n* = 3*x* = 24) cytotypes occur. In developmental terms, three main components of apomixis in *Taraxacum* can be distinguished: 1. diplospory—a modified form of meiosis in which chromosomal recombination and reduction are absent, resulting in non-haploid female gametophyte cells; 2. parthenogenesis—the direct development of an embryo from an egg cell without fertilization; and 3. the autonomous development of endosperm—endosperm arising from the dividing, hexaploid central cell of the female gametophyte. The need for an endosperm may at first appear non-obvious; however, the endosperm is a tissue that nourishes the developing embryo; without it, the embryo will abort [7]. In diploid sexual flowering plants, the triploid endosperm develops from the fertilization of the diploid central cell by a haploid sperm cell deposited by the pollen tube. Many apomicts also require the fertilization of the central cell to trigger the development of the endosperm (pseudogamy), making autonomous endosperm formation dispensable in these species. However, most, if not all, apomictic *Asteraceae* species produce autonomous endosperm [8].

With an understanding of the three developmental components of apomixis, the genes responsible need to be cloned in order to enable the introduction of apomixis into sexual crops, for which genetic analysis is necessary so that the number and complexity of the genetic loci underlying the developmental components are understood [2,9,10]. For example, are the developmental components controlled by separate, unlinked loci; is there a single chromosomal cluster of genes encoding different components; or is there a primary gene with different pleiotropic effects? Because apomicts undergo pollen meiosis, they can be used to investigate the genetics of apomixis in crosses with sexual seed plants, which has been used in the investigation of apomixis from two other genera of the *Asteraceae*. Based on various crosses between sexuals and apomicts of *Erigeron annuus* (fleabane), Noyes and colleagues proposed that that apomixis was controlled by a diplospory locus (*D*), whereas parthenogenesis and autonomous endosperm were controlled by a single fertilization factor locus (termed *F*, [11,12]). Similarly, deletion mutagenesis in *Hieracium* subgenus *Pilosella* (hawkweed) identified loci containing two of the components of apomixis: the locus for apospory, termed *Loss-of-Apospory* (*LOA*), and a locus containing the genes for both parthenogenesis and autonomous endosperm, termed *Loss-of-Parthenogenesis* (*LOP)*; [13,14]. However, further investigation of the *LOP* locus uncovered rare recombination events between the parthenogenesis phenotype and the autonomous endosperm phenotype, consistent with two tightly linked genes (*LOP* and Autonomous Endosperm, abbreviated as *AutE*). Unexpectedly, the genetic fine mapping of *AutE* in more crosses showed that this locus was on a different linkage group from the *LOP* locus, with a low penetrance (18%) and no additive effect [15], suggesting that the *AutE* phenotype in *Hieracium* is controlled by a more complex genetic mechanism than *LOA* or *LOP*.

Parthenogenesis has had extensive research focus in an attempt to understand the genes that drive the phenotype, both in natural apomicts as well as in model sexual systems (extensively reviewed in [16]), culminating in the cloning of the parthenogenesis-inducing transcription factor *BABY BOOM-Like* (*PsBBL*) from the monocot apomict *Pennisetum squamulatum* [17]. No other functionally validated parthenogenesis-controlling genes from apomicts have been described to date, so it is unclear if *PsBBL* is a universal mechanism for parthenogenesis in apomicts. By analysis of mutations in *Arabidopsis thaliana*, parthenogenesis was reported in mutants in the Polycomb Group 2 (PcG2) complex member *MULTICOPY SUPPRESSOR OF IRA1* (*MSI1*) [18] and in lines recovered from genetic screens performed by Fenby and colleagues [19].

Unlike parthenogenesis, the genetic control of the autonomous endosperm component of apomixis has not been thoroughly investigated outside of the apomicts in *Erigeron* and *Hieracium*. This is in part because many well-studied apomicts, such as *Pennisetum*, use pseudogamy and thus lack this component. Additionally, if present, the penetrance of the phenotype can be low, with *Boechera holboelli* showing a maximum penetrance of just 15% [20]. In contrast, the penetrance of autonomous endosperm development in *Taraxacum* is complete, making this system ideal for uncovering the basis of this component of apomixis, in combination with the wealth of information known about endosperm formation in general, as well as specific mutations that can induce its autonomous formation.

In *Arabidopsis*, mutations that give a Fertilization Independent Seed (FIS)-formation phenotype are found in genes encoding the PcG2 complex and act by initiating autonomous endosperm development, although the endosperm fails to cellularize and, eventually, the seeds abort development [21,22,23]. Therefore, *FIS* orthologs would appear to be clear candidate genes for autonomous endosperm development in autonomous apomicts and potentially more, as *MSI1* affects parthenogenesis as well. However, while the silencing of *FIS*-genes by RNAi in *Hieracium* affected endosperm development in sexuals, the transgene by itself did not induce autonomous endosperm [24], suggesting that alterations in the *FIS* genes may not be the causative lesions behind the autonomous endosperm formation in apomicts of the *Asteraceae*. In addition, the *Hieracium MSI1* gene did not map to the *LOP* locus, eliminating this gene as a candidate for parthenogenesis and autonomous endosperm [25]. Thus, one way to get insights into autonomous endosperm is to find the causative genes from apomicts such as *Taraxacum*.

Earlier studies in *Taraxacum* have focused on the inheritance of diplospory by using a tetraploid diplospory pollen donor [26]. The tetraploid diplospory pollen donor lacked parthenogenesis and was derived from a cross between a sexual seed plant and an apomictic pollen donor [27]. Co-dominant simple-sequence repeat (SSR) markers genetically linked to the diplospory-encoding *DIP* locus showed that this dominant component of apomixis was tetrasomically inherited, and the genotype had the *Dddd* simplex constitution [26]. Additionally, the *DIP* locus was weakly linked to the *18S–25S rDNA* locus, which was later confirmed by the fluorescent in situ hybridization of Bacterial Artificial Chromosome (BAC) probes on one of the Nucleolus Organizer Region (NOR) chromosomes [28]. Vijverberg and colleagues [29] used the tetraploid diplospory pollen donor described above for the genetic fine-mapping of the *DIP* locus and estimated the distance between the SSR *MSTA78-a* allele and the *DIP* locus to be 3.5 centimorgans (cM). Positional information of the *DIP* locus was used to identify the *DIP* gene as a *Vacuolar Protein Sorting-associated 13* (*VPS13*)-like gene [30]. The molecular function of this gene in apomixis is currently under investigation.

The fact that the tetraploid pollen donor plant above was diplosporous, but lacked parthenogenesis, suggested that diplospory and the parthenogenesis-encoding locus (*PAR*) in *Taraxacum* were controlled by at least two separate genetic loci. Van Dijk and colleagues [31,32] further investigated the breakdown of apomixis into its developmental components in non-apomictic offspring from crosses between diploid sexuals and triploid apomicts. Using Nomarski Differential Interference Contrast (DIC) microscopy of cleared ovules and SSR-marker analysis of progeny after pollination with diploid sexuals, they distinguished three non-apomictic phenotypes: type A, lacking diplospory and lacking parthenogenesis; type B, with diplospory and incomplete penetrance of parthenogenesis; and type C, with diplospory and autonomous endosperm but lacking parthenogenesis. Type B did not set seed in isolation, and because no autonomous endosperm development was seen with DIC microscopy, it was speculated that this type would need fertilization for endosperm development (pseudogamy). Mártonfiová et al. [33] found in crosses between diploid sexuals and tetraploid apomicts both type A and type C among the non-apomictic offspring but no type B. These studies suggest that apomixis in *Taraxacum* could be controlled by two or three major loci. However, the numbers of offspring in these crosses were too small for detailed segregation analysis.

To find the genes controlling parthenogenesis and autonomous endosperm in *Taraxacum*, the first question of if autonomous endosperm and parthenogenesis always co-occur and are controlled by a common dominant genetic factor needs to be addressed. As outlined in Figure 1, the inheritance (following a cross between a diploid sexual seed plant and an apomictic triploid pollen donor plant) of functional apomixis as a phenotype, along with the inheritance of the three components of apomixis, can be conceived in two alternative hypotheses: as either two or three unlinked genetic loci. In the two-locus model, parthenogenesis and autonomous endosperm formation are determined by a single common locus (such as the *F* locus of *Erigeron*); in the three-locus-model, each apomictic component is determined by a separate unlinked locus. Assuming that the loci are uncoupled and have the same genotypic constitution as the diplospory locus, 44 percent of the offspring from the two-locus model are expected to be apomictic, while in the three-locus model, 30 percent of the offspring are expected to be apomictic.

In this article, the inheritance of the individual components of apomixis in a cross between a diploid sexual plant and an apomictic triploid plant is described. Although pollen fertility due to triploidy was very low, by making a large number of crosses, it was possible to generate sufficient triploid offspring, allowing the quantification of the segregation of different elements of apomixis. As previously found, the SSR *MSTA78-a* allele was tightly linked to diplospory. This allele was used as a molecular marker for diplospory so that the phenotyping could focus on elucidating parthenogenesis and autonomous endosperm formation in the diplosporous progeny. It is critical to focus on the diplosporous progeny because both parthenogenesis and autonomous endosperm are gametophytic traits, which will segregate on a plant with normal meiosis. Unfortunately, meiosis will be highly disturbed due to the chromosomal imbalance inherent in the triploid F_1_ plants, causing the seed phenotypes to show segregation distortion, which impedes clear analysis. When focusing the analysis on the diplosporous progeny, the problem of segregation distortion is avoided because these F_1_ plants do not undergo meiosis. Analysis of the F_1_ showed that the SSR *MSTA44B-d* allele is tightly linked to parthenogenesis: the first molecular marker for parthenogenesis in *Taraxacum*. Additionally, while 29% of the diplosporous offspring contained all three of the components of apomixis, which is consistent with the three-locus model, nearly 95% of the diplosporous offspring showed some degree of autonomous endosperm formation, independent of whether or not the egg cell underwent parthenogenesis. The absence of the clear segregation of autonomous endosperm development in *Taraxacum* is different from the coupling between autonomous endosperm and parthenogenesis previously reported in other apomictic *Asteraceae*.

## 2. Materials and Methods

### 2.1. The Cross between a Sexual Diploid and an Apomictic Triploid

Pollen from a pollen-fertile apomictic *T. officinale* line triploid A68 was crossed with stigmas from the pollen-sterile sexual diploid *T. officinale* line TJX3-20 [34]. TJX3-20 originated from Langres in France, while A68 originated from Heteren in the Netherlands. TJX3-20 produces small, infertile pollen due to a cytoplasmic male sterility system [35]. The seed parent’s pollen infertility avoids the breakdown of the sporophytic Self Incompatibility System, which occurs in crosses when triploids are used as pollen donors and would generate many diploid selfed offspring (“mentor pollen effect” [36]). From a total of 62 crossed capitula (inflorescences), only 192 viable seeds were produced in total. The F_1_ seeds were sown, and the resulting F_1_ plants were screened by flow cytometry to determine ploidy.

### 2.2. Apomixis Phenotyping

To induce flowering, eight-week-old F_1_ plants were vernalized for nine weeks in a cold room at 4 °C. Like the seed parent TJX3-20, all the F_1_ plants were male sterile and physically isolated from other wild-type *Taraxacum*, eliminating the possibility of sexual seed setting. Under these conditions, the development of a large seed head is a sign of apomictic seed setting. To determine the degree of apomictic seed setting, for each F_1_ plant, 50 randomly chosen brown seeds were germinated, and the numbers of seedlings germinating were counted. Germinating seeds implied that the mother was apomictic and had all components of apomixis.

### 2.3. Microsatellite Genotyping

Twelve codominant SSR loci were screened (Micro Satellite TAraxacum (MSTA) [37,38]: *MSTA31*, *MSTA44B*, *MSTA53*, *MSTA64*, *MSTA67*, *MSTA73*, *MSTA74*, *MSTA78*, *MSTA85*, *MSTA101*, *MSTA105*, and *MSTA131* (Table 1). DNA extraction and SSR assays were performed as described in [26]. The PCR products were analyzed on an ALF express II automatic sequencer (Amersham, Pharmacia Biotech, Charfonte, UK).

### 2.4. Nomarski DIC Microscopy Phenotyping

Methyl-salicylate cleared ovules from plants that were *MSTA78-a* positive but did not show seed setting (i.e., were apomixis-negative) were investigated by Nomarski DIC microscopy, as described in [39]. At least ten different ovules were analyzed per plant at anthesis or one day after anthesis.

### 2.5. Seed Flow Cytometry

Matzk and colleagues [40] have shown that autonomous endosperm production in ripe 3 × *Taraxacum* apomictic seeds can be detected with a flow cytometer as a 6 × peak, derived from the unfertilized central cell, which contains two fused 3 × polar nuclei (flow cytometric seed screen, FCSS). Matzk and colleagues also reported that endopolyploidization peaks were absent in the autonomous apomictic seeds of *T. officinale* and *Hieracium pilosella*, suggesting that the presence of 6 × peaks in isolated seeds would be due to autonomous endosperm formation. To assess the ploidy of the endosperm, developing seeds from the apomictic F_1_ plants were collected five days after anthesis and directly analyzed by flow cytometry. Ten developing seeds were homogenized in Otto I buffer [41] by chopping the seeds with a sharp razor blade. Ploidy levels were determined with a flow cytometer (Ploidy analyzer, Partec, Münster, Germany) using 4′, 6-diamidino-2-phenylindole (DAPI) as a fluorescent stain as described in [36].

## 3. Results

### 3.1. Segregation of Apomixis as a Whole

As previously described by Van Dijk and colleagues [34], stigmas from a pollen-sterile sexual diploid dandelion (TJX3-20) were crossed with pollen from a pollen-fertile apomictic triploid (A68). From a total of sixty-two crossed capitula (inflorescences, see Figure 2A), only 192 viable F_1_ seeds were produced in total.

The average seed set per seed head from this cross was only 2.1 seeds, while, in the diploid × diploid sexual crosses, an average of ~100 seeds per seed head were obtained. Flow cytometry showed that 96 of the 192 F_1_ plants were diploid (50%), 95 were triploid (49.5%), and one (0.5%) was tetraploid. These plants were the products of the fertilization of a haploid egg cell by a haploid, diploid, or triploid pollen grain, respectively. A68 produced many collapsed and small pollen grains, reflecting the high frequency of inviable aneuploid pollen grains produced because of unbalanced triploid pollen meiosis. However, no viable aneuploid offspring was recovered, which was fortunate, as the lack of aneuploidy in the F_1_ simplifies the interpretation of the genetic analysis of apomixis. None of the diploid F_1_ plants were apomictic, while the triploid F_1_ plants segregated for apomixis. The only tetraploid plant recovered was also capable of apomictic reproduction.

To gain further insights into the genetics of apomixis, Van Dijk and colleagues [34] (2009) also reported on the transmission of the diplospory-linked marker *MSTA78-a* to diploid and triploid hybrids of this cross. While the segregation ratio of the three alleles in the triploid progeny was not significantly different from the Mendelian 1:1:1 ratio, the ratio was highly distorted in the diploid hybrids (0.01:0.47:0.52). Assuming that the single case of a *MSTA78-a* allele transmitted to a diploid hybrid was a recombination event between the *a*-allele and the *DIP* allele, it was postulated that haploid pollen grains carrying the diplospory allele were lethal [34]. To expand this analysis, the further investigation of the inheritance of the developmental components of apomixis in triploid progeny described here was undertaken.

As one triploid F_1_ plant died before its reproductive system could be assessed, the remaining 94 3 × plants were used for further analysis. Apomictic seed setting was variable, as can be seen in the examples shown in Figure 3. As shown in Figure 4, 69 percent of the triploid F_1_ plants produced no germinating seeds (64/93; for one apomictic F_1_ plant (H69), seed setting was not quantified). By contrast, 24 percent of the triploid F_1_ plants produced more than 80% germinating seeds (22/93). The remaining six plants produced seeds with a germination rate between 6 and 78 percent, which may be due to incomplete penetration of apomixis factors. The two-locus model outlined in Figure 1 predicts 41 of the 94 triploid offspring plants to be apomictic, while the three-locus model predicts a significantly fewer 28 apomicts. A total of twenty-nine triploid apomicts were recovered, fitting the three-locus model well (χ^2^ = 0.05, degrees of freedom (d.f.) = 1; the *p*-value is 0.82, not significant at *p* < 0.05) and contradicting the two-locus model (χ^2^ = 6.23, d.f. = 1; the *p*-value is 0.01, significant at *p* < 0.05).

### 3.2. Association between SSR Markers and Apomixis

To further understand the segregation of the components of apomixis, DNA from the progeny was screened with additional molecular markers (as described in [37,38]). In Table 2, the allele distributions of eleven SSR loci in the 29 apomictic F_1_ triploids and the 66 non-apomictic F_1_ triploids are listed. *MSTA31* could not be scored due to the presence of null alleles in the parental genotypes. As expected, all 29 F_1_ triploids that reproduced via apomixis carried the paternal *MSTA78-a* (random association: χ^2^ = 12.65; d.f. = 1; the *p*-value is 0.0004, significant at *p* < 0.05), supporting the previously reported tight linkage between the *MSTA78-a* allele and the *DIP* locus [26,29]. Due to this, the *MSTA78-a* SSR was used as a molecular marker for the presence of diplospory induced by the *DIP* locus (independent of the phenotypic assessment of apomixis) and will be referred to as the “Dip-marker” below. Molecular analysis showed that 63 of the 94 3 × F_1_ plants carried the *MSTA78-a* allele and were considered to be diplosporous. The observed frequency of 65 percent of F_1_ plants carrying the Dip-marker is also consistent with the expected two-thirds of the diploid A68 pollen grains carrying the *D* allele (χ^2^ = 0.19; d.f. = 1; the *p*-value is 0.66, not significant at *p* < 0.05). Due to the function of the *DIP* locus blocking meiosis, no gametophytic segregation of the parthenogenesis or autonomous endosperm loci is expected in these plants. To see if molecular markers for either parthenogenesis or autonomous endosperm formation could be uncovered, these F_1_ plants were genotyped for additional markers for linkage to the apomictic phenotype (as shown in Table 2). Of the remaining 10 SSR markers, one, *MSTA44B*, showed a significant association with apomixis. The genotype of the triploid pollen donor A68 for *MSTA44B* was *b* (167 bp)/*d* (171 bp)/*e* (183 bp).

Of the 29 apomictic F_1_ plants, 27 had the *d* allele and two did not (H23 and H133, see Table 3). Furthermore, the *MSTA44B-d* allele is unlinked to the Dip-marker, consistent with it being genetically unlinked to the *DIP* locus (random association: χ^2^ = 1.36; d.f. = 1; the *p*-value is 0.85, not significant at *p* < 0.05). To determine if the *MSTA44B-d* allele was linked to either parthenogenesis or autonomous endosperm formation, Nomarski DIC microscopy phenotyping of developing seeds was performed.

### 3.3. Nomarski DIC Microscopy Phenotyping of Parthenogenesis and Autonomous Endosperm

For the triploid F_1_ plants that produced viable seeds, it is reasonable to assume that all apomixis components are present (see Figure 5A, autonomous apomict H99). To further elucidate the impacts on seed development, developing seeds from several complete apomicts were cleared and imaged with Nomarski DIC microscopy for the presence of embryos and autonomous endosperm. The developing seeds of the F_1_ were highly asynchronous in parthenogenetic embryo development and autonomous endosperm formation. Figure 5B shows an embryo sac with an advanced embryo but a non-divided central cell nucleus of the non-apomictic plant H70. The hexaploid peak in the FCSS indicates that some level of autonomous endosperm developed later, but ultimately, no viable seeds were produced from this F_1_ plant. By contrast, Figure 5C shows an embryo sac with an undivided egg cell and an advanced cellularized endosperm; again, no viable seeds were produced from this F_1_ plant. As a result, it is possible that, at the time of fixation, the development of either has not yet started, which would provide an erroneous negative score. To account for the asynchrony, the presence of a multicellular embryo was scored as positive, while the presence of an undeveloped egg was scored as “inconclusive”. Similarly, the formation of endosperm was also scored as positive, while the presence of an undeveloped central cell was scored as “inconclusive”.

There were four non-apomictic F_1_ plants (H22, H56, H70, and H163) carrying the *MSTA44B-d* allele, in which parthenogenetic embryos were observed with DIC microscopy but that had no observed autonomous endosperm formation and no viable seed set (Figure 5D; H22). Interestingly, the *MSTA44B-d* allele was previously shown to be transmitted by haploid pollen grains to the diploid progeny (in contrast to the Dip-marker, [34]), although at a ratio significantly different from the expected Mendelian ratio (allele frequencies: b = 0.37; d = 0.17; e = 0.45 (*N* = 94); χ^2^ = 12.20; d.f. = 2; the *p*-value is 0.002, significant at *p* < 0.05). Additionally, in two diploid male-sterile progeny-plants carrying the *MSTA44B-d* allele, embryo-like structures, and autonomous endosperm-like tissues were visible with Nomarski DIC microscopy (see Figure 6). These diploid plants, however, did not produce viable seeds. Taken together, these lines of evidence suggest that this SSR allele is genetically linked to the parthenogenesis locus, and this allele will be referred to as the “Par-marker” below.

### 3.4. Seed Flow Cytometry to Assess Autonomous Endosperm Formation

Due to the previously mentioned asynchronous initiation of autonomous endosperm, in order to perform a more robust assessment of the autonomous endosperm phenotype, the F_2_ seeds of 24 3 × F_1_ plants were screened via flow cytometry (FCSS) to determine their ploidy (Table 3; Figure 7). By combining the number of apomictic 3 × F_1_ (29, which, by definition, have autonomous endosperm formation) along with the non-apomictic 3 × F_1_ that had the DIP-marker allele and in which autonomous endosperm was directly observed (24, by either Nomarski DIC microscopy or FCSS), a total of 53 diplosporous 3 × F_1_ plants demonstrated autonomous endosperm formation. Only three did not show any sign of autonomous endosperm formation, and the remaining seven could not be scored (due to either failure to induce flowering or premature death). According to the three-locus model with a single dominant autonomous endosperm formation locus (Figure 1), only 38 plants with autonomous endosperm are expected (67 percent of 56; goodness-of-fit χ^2^ = 18.42; d.f.= 1; the *p*-value is 0.00, significant at *p* < 0.05). Additionally, for the 20 diplosporous (Dip-marker present) 3 × F_1_ plants that lacked the parthenogenesis-linked Par-marker allele, 19 3 × F_1_ plants showed autonomous endosperm formation (95 percent). Thus, significantly more plants with autonomous endosperm were found than were expected, which suggests that the developmental function of autonomous endosperm formation in apomixis is more complex in *Taraxacum* than can be determined by a single dominant genetic locus.

## 4. Discussion

Apomixis has since long ago been a holy grail of agriculture [3,4]; now, over a quarter of a century later, the molecular nature of the genes driving apomixis is starting to come into view, largely due to the advances in apomictic model systems such as *Hieracium, Pennisetum*, and *Taraxacum*, starting with a fulsome understanding of the underlying genetic system [42,43]. The progeny of the sexual x apomict *Taraxacum* cross examined here segregated for apomixis as a whole trait as well as for the individual components of apomixis. When considered as a single trait, apomixis was consistent with a three-locus model controlling apomixis. In addition to the already-known SSR marker for diplospory (*MSTA78-a*; Dip-marker), a co-dominant marker tightly linked to the parthenogenesis allele was found (*MSTA44B-d*; Par-marker) that was not genetically linked to the Dip-marker. Thus, the genotype for parthenogenesis is dominant simplex *Ppp*, and the parthenogenesis and diplospory loci are not genetically linked. A dominant simplex parthenogenesis genotype has also been reported in other members of the *Asteraceae*, such as *Erigeron* [11,12] and *Hieracium* [13,14], perhaps with broader developmental roles then shown here for *Taraxacum*. The presence of a single, major parthenogenesis locus significantly enables the cloning of the parthenogenesis gene in *Taraxacum*, as was done for the *VPS13-like* gene in the *DIPLOSPOROUS* locus [30]. Reducing the number of potential apomixis genes in a forward genetic mutation screen or a comparative transcriptomic study to a small set of positional candidates significantly increases the chances of finding causal apomixis genes.

While having the Par-marker will aid in finding the causal gene, the introduction of apomixis into crop plants also requires endosperm formation: either pseudogamy or autonomous formation. In this study, the segregation of autonomous endosperm was investigated in the segregating 3 × F_1_ through a combination of cytological observations, seed flow cytometry, and molecular marker analysis. That autonomous endosperm did not appear to segregate was unexpected, because, in an earlier cytological study, phenotypically complementary recombinants between parthenogenesis and autonomous endosperm had been found [32]. An explanation could be that parthenogenesis and/or autonomous endosperm developed late and that the cytological observations were performed too early for parthenogenesis or autonomous endosperm development to become expressed. Fagerlind [44] found a wide variation in developmental stages between florets within a capitulum (inflorescence) of a completely apomictic *T. officinale*. In addition, Cooper and Brink [45] described a high degree of independent development of the embryo and endosperm within florets in a completely apomictic *T. officinale*, while development was very synchronous in the sexual diploid related species *T. koksaghyz*. As extreme examples for *T. officinale*, Cooper and Brink [45] describe a unicellular endosperm (central cell) with a 112-celled embryo and, conversely, an egg cell with a 128-celled endosperm. An advantage of FCSS in this regard is that it pools several developing seeds (hence leveling out developmental variation between florets that is genotype independent) and that the ploidy level indicates whether there is autonomous endosperm (6 × peak). In the present study, microscopic observations in combination with seed flow cytometry of developing seeds five days after flowering suggests that the autonomous endosperm does not segregate in Dip-marker positive triploid progeny plants. The three plants in which no autonomous endosperm could be demonstrated may be explained by recombination between the Dip-marker and the *Diplosporous* locus/gene, because at a distance of 3.5 cM [29], 2–3 meiotic plants are expected among 62 Dip-marker positive plants. An alternative explanation for the failure to find the expected segregation of autonomous endosperm in these experiments could be that the locus for autonomous endosperm is genetically closely linked to the diplospory locus, as functional diplospory was assessed via a linked Dip-marker. In this case, the autonomous endosperm phenotype would be “fixed” in the 61 3 × F_1_ lines used for analysis or, at most, ~5 cM away. However, there is no sign of the suppression of recombination in the diplospory locus [27,29], making this explanation unlikely. Lastly, it is possible that the apomictic pollen donor A68 used in these experiments was homozygous for genetic autonomous endosperm factors or that autonomous endosperm is not genetically determined but is, for example, a direct consequence of polyploidy. If autonomous endosperm formation is fixed in this population, then a developmental function needs to be assigned to the third segregating dominant apomixis locus, yet to be identified.

Our results strongly suggest an independent control of autonomous endosperm and parthenogenesis in *T. officinale*. This is supported by the study of Mártonfiová and colleagues [33]. They investigated the components of apomixis in 3 × progeny of a sexual x apomictic cross, using test crosses with a diploid sexual pollinator and FCSS of the developing seeds. Three plants were found that had lost parthenogenesis but made hexaploid autonomous endosperm, showing that parthenogenesis and autonomous endosperm formation do not have a common genetic control. These observations in *Taraxacum* are clearly different from what was seen in *Erigeron* and *Hieracium*. In these other apomictic *Asteraceae* species, autonomous endosperm co-segregated with parthenogenesis, either due to genetically linked genes in a single locus or due to the pleiotropic effects of a single gene. Since the activation of the egg to develop into an embryo and activation of the central cell to develop into an autonomous endosperm are in some ways similar processes, pleiotropy is a possibility, consistent with the observations of Guitton and Berger [18] in *Arabidopsis*. It is also possible that in some cases, at least, the parthenogenetic embryo triggers autonomous endosperm development, as has been reported in *Arabidopsis* for a mutant with a single sperm cell [46].

To aid the isolation of apomixis components, Van Dijk and colleagues [30] generated a γ radiation deletion population of A68, the same apomictic clone used in the crossing described here. In addition to mutants for the diplospory locus, mutants for the parthenogenesis locus were found (Van Dijk et al., in preparation). That these mutants had lost the dominant parthenogenesis allele but still had the dominant diplospory allele was shown by the fact that test crosses with a diploid pollen donor only produced tetraploid progeny. However, these PAR deletion mutants still made autonomous endosperm, as shown by FCSS and Nomarski DIC microscopy. This is in line with the crossing results that parthenogenesis and autonomous endosperm are independently genetically regulated and again underscores the difference in mechanism between *Taraxacum* and other members of the *Asteraceae*.

## 5. Conclusions

Diplospory and parthenogenesis are clearly major loci of apomixis in *Taraxacum*, each of which is controlled by a single dominant locus. However, apomixis as a whole does not fit a two-factor model, suggesting the presence of a third, critical dominant locus. Based on the observations presented here, this locus does not appear to be the single dominant autonomous endosperm formation locus that might have been expected. Apomixis may be co-controlled by a third unknown major locus of unknown developmental function or by several smaller background or modifier genes that jointly appear as a third main factor in this cross. In order to distinguish between these various hypotheses, further research is needed, which is critical if there is to be any progress in cloning the genes involved in the autonomous endosperm formation component of apomixis. While autonomous endosperm is a major component of apomixis in the *Asteraceae*, it is still the least understood.

## Figures and Tables

**Figure 1 genes-11-00961-f001:**
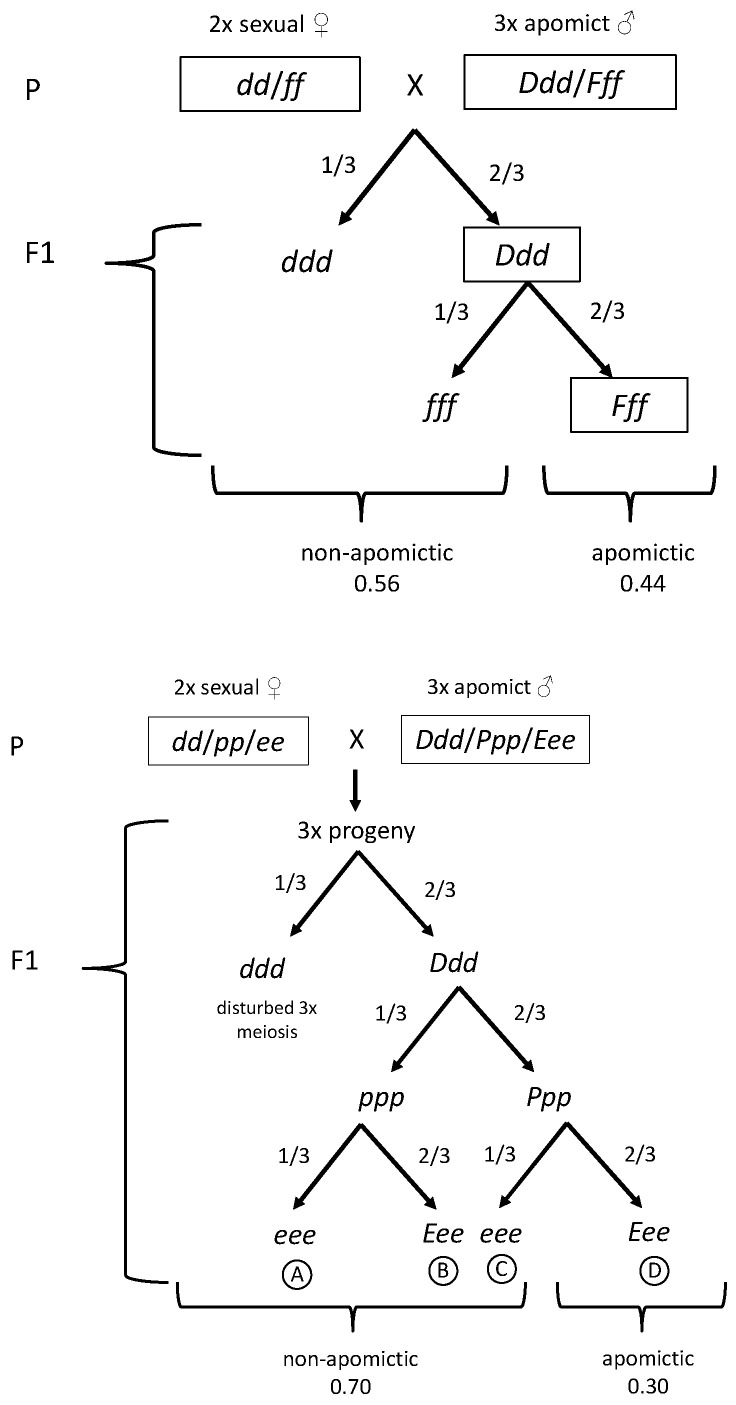
Two alternative genetic models for the polysomic inheritance of apomixis in *Taraxacum*. Both start with a cross between a diploid seed plant and a triploid apomictic pollen donor, and only triploid offspring are shown (produced by diploid pollen grains). The top panel shows a two-locus model with one factor *F* controlling both parthenogenesis and autonomous endosperm development. The bottom panel shows a three-locus model with separate factors for parthenogenesis and autonomous endosperm development. *D* is diplospory, *d* is meiotic, *F* is the fertilization factor, *f* is fertilization-dependent, *P* is parthenogenesis, *p* is fertilization-dependent, *E* is autonomous endosperm, and *e* is sexual endosperm.

**Figure 2 genes-11-00961-f002:**
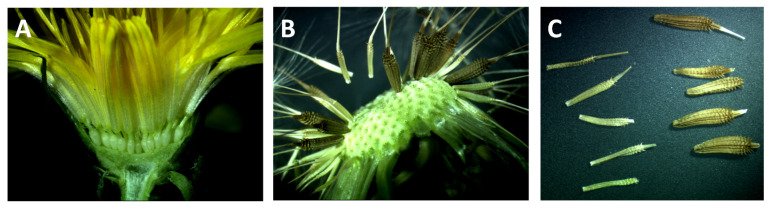
(**A**) A longitudinal section of a capitulum (inflorescence) of *Taraxacum officinale*, showing the florets attached to the receptacle. The inferior ovaries have a single ovule and produce a single-seeded fruit (achene), often referred to as “seed”. (**B**) An opened seed head of an apomict that produces both light and dark brown seeds (about ten days after flowering). Light brown seeds are empty ((**C**), left), while dark brown seeds ((**C**), right) generally germinate.

**Figure 3 genes-11-00961-f003:**
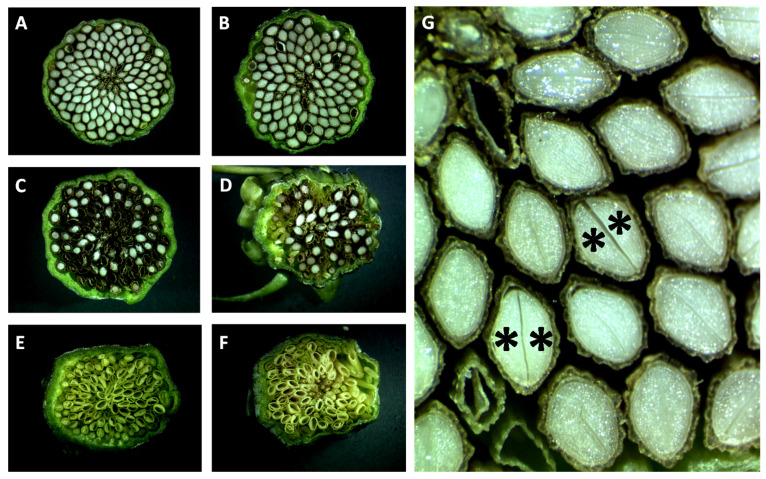
Variation in apomictic seed setting of F_1_ triploids. Cross sections of seeds in (closed) seed heads of some 3 × hybrid progeny plants, shortly before opening and seed shatter (about nine days after anthesis). (**A**,**B**) are apomicts with high penetrance of apomixis (H2 and H18); (**C**,**D**) are apomicts with low penetrance (H37 and H99); (**E**,**F**) are two non-apomicts (H70 and H86). At this stage, the viable seeds are mainly filled with the two cotyledons (asterisks, panel (**G**)). The apomicts with low penetrance produce a seed coat, even in empty seeds; the non-apomicts produce no seed coat.

**Figure 4 genes-11-00961-f004:**
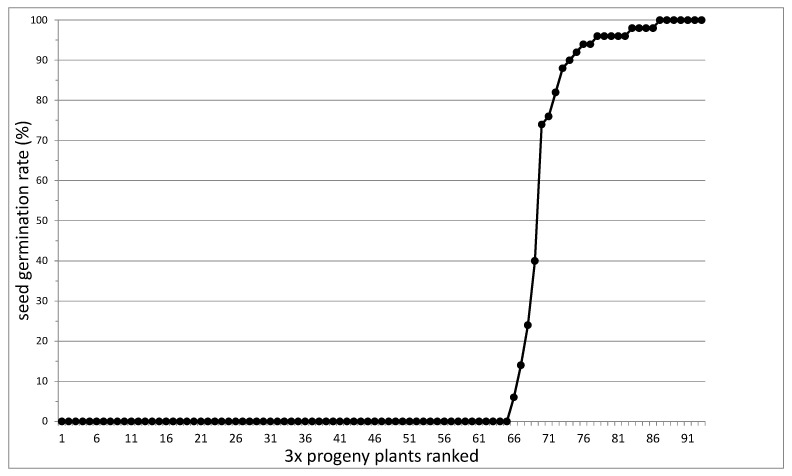
The germination rate of 93 3 × progeny plants.

**Figure 5 genes-11-00961-f005:**
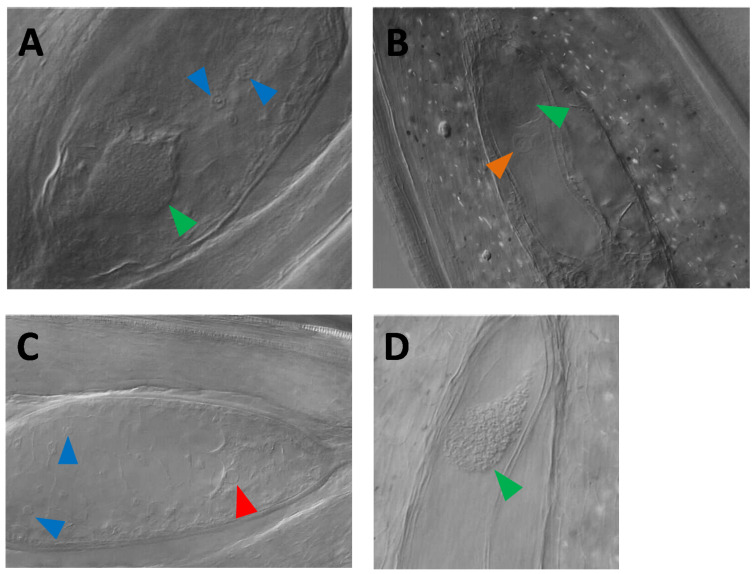
Examples of embryo sac development in 3 × plants. (**A**) A developing seed with a parthenogenetic embryo and autonomous endosperm from the apomictic plant H99. Globular embryo (green arrow) and endosperm nuclei (blue arrows) are visible. (**B**) A developing seed with a multicellular embryo (green arrow) with an undivided central cell of the gametophyte (orange arrow) from the non-apomictic plant H70. The formation of endosperm was seen in this line via FCSS as a hexaploid peak, showing autonomous endosperm was initiated at a later stage. (**C**) A developing seed without parthenogenesis but with cellularized autonomous endosperm development from the non-apomictic plant H123. The red arrow points to the egg cell nucleus; the blue arrows, to endosperm nuclei. (**D**) A developing seed with a parthenogenetic embryo (green arrow) with no detectable autonomous endosperm formation from the non-apomictic plant H22, which carries the Par-marker. Since no FCSS data were available for this line, the information about autonomous endosperm development is non-conclusive.

**Figure 6 genes-11-00961-f006:**
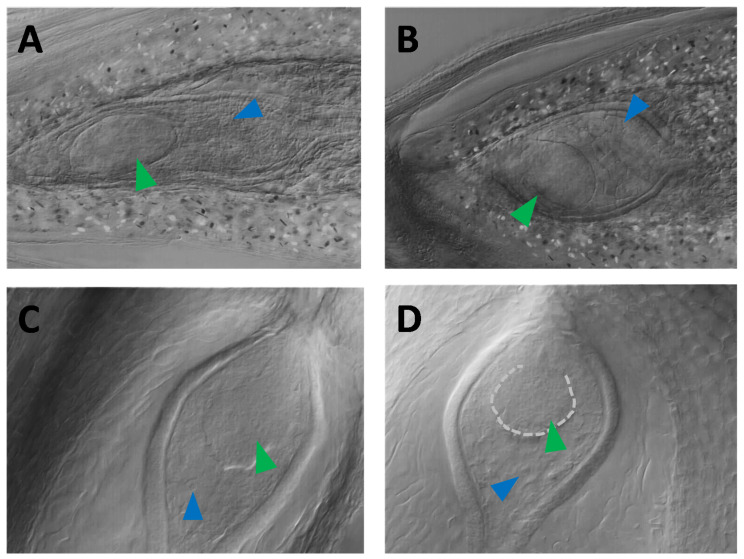
Examples of parthenogenetic embryo-like structures in 2 × non-apomictic plants that carry the Par-marker. (**A**) and (**B**): plant H47; panels (**C**) and (**D**): plant H17. The green arrows point to the embryo-like structures; the blue arrows, to the endosperm nuclei/cells.

**Figure 7 genes-11-00961-f007:**
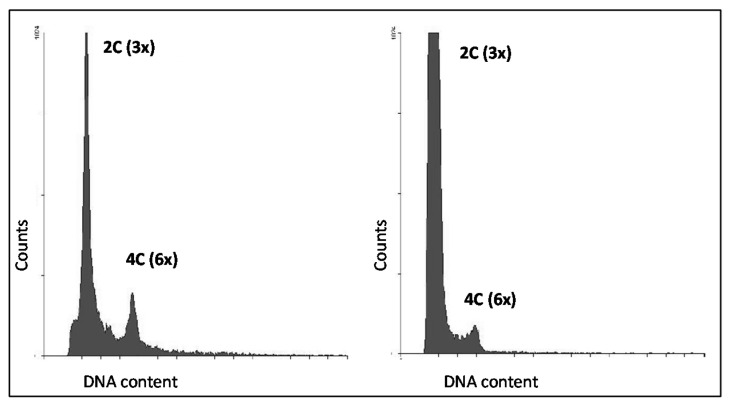
Examples of the Flow Cytometric Seed Screen. (**Left**): An apomictic triploid F_1_ plant, with a triploid somatic (maternal and embryo tissue) and a hexaploid endosperm peak showing autonomous endosperm. (**Right**): A non-apomictic triploid F_1_ plant, with a triploid somatic (maternal tissue) and a lower hexaploid endosperm peak, also showing the presence of autonomous endosperm.

**Table 1 genes-11-00961-t001:** Primer sequences of twelve codominant simple-sequence repeat (SSR) markers used in this study. For each primer pair, the repeat motif, primer sequences, optimal annealing temperature (TA, in °C) and expected size (in bp) are indicated, along with the reference where the SSR marker was first described.

Marker	Repeat Motif	Forward Primer Sequence	Reverse Primer Sequence	T_A_	Size	Reference
MSTA31	(CT)_17_	CCTCAAAGCCCGAACTT	ACGACCCCAACTGATTTTTAC	51.0	240	[37]
MSTA44B	(CT)_19_	AGTTTCTCTAAAATGGGAAGAT	TGTCAGGTATATTCAAAAGATTC	51.0	191	[37]
MSTA53	(TC)_12_(GT)_8_	CAATTATTATGGTCTCGTCCTT	CCAGTTGAAGCAAAAACAGT	55.0	203	[37]
MSTA64	(TC)_4_TT(TC)_2_TT(TC)_2_TT(TC)_5_A(TC)_4_-(A)_16_	TGCTTTTTGAACGACAGTG	TTTGCTTGGTTATTAGTGAACAT	55.0	191	[37]
MSTA67	(TC)_22_T(CA)_12_	TTCGGATATGACCCTTCACT	GACATCTTGCACCTAAAAACAAT	56.0	219	[37]
MSTA73	(TC)_21_CTG(TC)_8_	CCGCATGAGGTTGTCT	TGGGCTGTTTAATAGAACTTA	53.0	216	[37]
MSTA74	(CT)_10_	GAGGTCTTTTTATTCGGTTTT	GGATGCCTTTACAGTTACAAT	49.0	223	[37]
MSTA78	(CT)_9_	TGATTGATTCTGCCCTAAACC	TGCCAAGACATCCGAAAAG	52.0	151	[37]
MSTA85	(CT)_20_	TGCATGTTCGTTCTACTGGT	ACGTAATAAAATTGGAAGTCAGG	55.0	196	[37]
MSTA101	(CCT)_2_TCT(TC)_16_	GCATGGGGGTCGAGGGGTAT	CCGCGATGGACTTATTCTTGGTTG	57.8	198	[38]
MSTA105	(TC)_23_	CACCGTTCAAAAATAAAGATAAAA	AGAATAGCTCCGTCAAGTAGG	54.3	203	[38]
MSTA131	(AT)_7_	TACCCTGCAAACATTACTCTTCTG	GTTGGCCTGTTAATACTTGATACG	55.0	181	[38]

**Table 2 genes-11-00961-t002:** Allele frequencies of 11 paternal A68 SSR markers in apomictically reproducing progeny and non-apomictically reproducing progeny. A68 is a triploid and carries two or three different alleles per locus, which are indicated by a different letter (a–e). The *MSTA44B-d* (Par-marker) and the *MSTA78-a* (Dip-marker) alleles are significantly overrepresented in the apomicts.

SSR Locus	Allele 1	Allele 2	Allele 3	Chi-Square	d.f.	*p*-Value	Significance
*MSTA44B*	*b*	*d*	*e*				
apomicts	20	27	11				
non-apomicts	47	31	50	11.36	2	0.00	*p* < 0.05
*MSTA53*	*a*	*b*	*c*				
apomicts	14	20	16				
non-apomicts	44	44	38	0.81	2	0.67	n.s.
*MSTA64*	*a*	*b*	*c*				
apomicts	17	14	17				
non-apomicts	37	36	41	0.55	2	0.93	n.s.
*MSTA67*	*a*	*d*	*e*				
apomicts	24	17	18				
non-apomicts	38	46	46	2.43	2	0.30	n.s.
*MSTA73*	*a*	*a*	*c*				
apomicts	19		9				
non-apomicts	38		19	0.01	1	0.91	n.s.
*MSTA74*	*b*	*c*	*e*				
apomicts	22	17	15				
non-apomicts	44	34	46	1.45	2	0.48	n.s.
*MSTA78*	*a*	*c*	*d*				
apomicts	29	18	11				
non-apomicts	33	46	43	9.98	2	0.01	*p* < 0.05
*MSTA85*	*a*	*b*	*b*				
apomicts	20	36					
non-apomicts	32	90		1.67	1	0.20	n.s.
*MSTA101*	*a*	*c*	*d*				
apomicts	27	12	15				
non-apomicts	51	36	25	1.84	2	0.40	n.s.
*MSTA105*	*b*	*c*	*d*				
apomicts	17	18	21				
non-apomicts	36	44	42	0.28	2	0.87	n.s.
*MSTA131*	*a*	*c*	*d*				
apomicts	13	21	24				
non-apomicts	44	41	45	2.49	2	0.29	n.s.

**Table 3 genes-11-00961-t003:** Analysis of apomixis and its developmental components in 62 Dip-marker positive 3 × F_1_ plants.

	Ploidy	Germin. %	Apomixis	Dip Marker (*MST78-a*)	Par Marker (*MST44B-d*)	DIC Microscopy			
						PAR	AUT	FCSS AUT	AUT Combined
TJX 320	2 ×		no	ab	ac	non	non	non	–
68	3 ×	98	yes	acd	bde	yes	yes	yes	+
2	3 ×	96	yes	ac	bd	n.d.	n.d.	n.d.	+
16	3 ×	100	yes	ac	bd	n.d.	n.d.	n.d.	+
18	3 ×	96	yes	ad	bd	n.d.	n.d.	n.d.	+
23	3 ×	14	yes	ac	be	n.d.	?	n.d.	+
24	3 ×	100	yes	ad	bd	n.d.	?	n.d.	+
31	3 ×	100	yes	ad	bd	n.d.	n.d.	n.d.	+
34	3 ×	98	yes	ac	bd	n.d.	n.d.	n.d.	+
37	3 ×	40	yes	ac	bd	+	+	n.d.	+
48	3 ×	78	yes	ac	bd	+	+	n.d.	+
50	3 ×	98	yes	ac	bd	n.d.	n.d.	n.d.	+
69	3 ×	n.q.	yes	ad	de	+	+	n.d.	+
73	3 ×	100	yes	ac	de	+	+	n.d.	+
76	3 ×	94	yes	ad	bd	n.d.	n.d.	n.d.	+
98	3 ×	100	yes	ac	bd	n.d.	n.d.	n.d.	+
99	3 ×	74	yes	ad	bd	+	+	n.d.	+
113	3 ×	90	yes	ad	bd	n.d.	n.d.	n.d.	+
115	3 ×	96	yes	ad	de	n.d.	n.d.	n.d.	+
127	3 ×	82	yes	ad	bd	+	+	n.d.	+
132	3 ×	94	yes	ac	bd	n.d.	n.d.	n.d.	+
133	3 ×	98	yes	ac	be	n.d.	n.d.	n.d.	+
136	3 ×	96	yes	ac	de	n.d.	n.d.	n.d.	+
144	3 ×	88	yes	ac	de	n.d.	n.d.	n.d.	+
154	3 ×	100	yes	ac	bd	n.d.	n.d.	n.d.	+
158	3 ×	96	yes	ac	de	n.d.	n.d.	n.d.	+
159	3 ×	24	yes	ac	de	n.d.	n.d.	n.d.	+
165	3 ×	98	yes	ac	bd	n.d.	n.d.	n.d.	+
183	3 ×	92	yes	ac	bd	n.d.	n.d.	n.d.	+
194	3 ×	6	yes	ad	de	n.d.	n.d.	n.d.	+
201	3 ×	100	yes	ad	de	n.d.	n.d.	n.d.	+
163	3 ×	0	no	ad	de	+	+	+	+
70	3 ×	0	no	ac	bd	+	?	+	+
22	3 ×	0	no	ac	de	+	?	n.d.	?
56	3 ×	0	no	ac	de	+	?	n.d.	?
148	3 ×	0	no	ac	de	?	+	+	+
95	3 ×	0	no	ac	bd	?	?	+	+
114	3 ×	0	no	ac	de	n.d.	n.d.	+	+
123	3 ×	0	no	ad	de	?	+	n.d.	+
195	3 ×	0	no	ac	de	?	?	+	+
175	3 ×	0	no	ac	bd	n.d.	n.d.	–	–
193	3 ×	0	no	ac	de	n.d.	n.d.	–	–
30	3 ×	0	no	ad	be	?	+	+	+
45	3 ×	0	no	ad	be	?	+	+	+
139	3 ×	0	no	ad	be	?	?	+	+
170	3 ×	0	no	ad	be	?	+	+	+
181	3 ×	0	no	ac	be	?	+	+	+
185	3 ×	0	no	ac	be	?	+	+	+
190	3 ×	0	no	ac	be	?	+	+	+
177	3 ×	0	no	ac	be	?	+	n.d.	+
68	3 ×	0	no	ad	be	n.d.	n.d.	+	+
78	3 ×	0	no	ad	be	n.d.	n.d.	+	+
79	3 ×	0	no	ac	be	?	?	+	+
86	3 ×	0	no	ad	be	n.d.	n.d.	+	+
117	3 ×	0	no	ad	be	n.d.	n.d.	+	+
140	3 ×	0	no	ad	be	?	?	+	+
149	3 ×	0	no	ac	be	?	?	+	+
176	3 ×	0	no	ad	be	?	?	+	+
207	3 ×	0	no	ad	be	?	?	+	+
65	3 ×	0	no	ac	be	?	?	–	–
112	3 ×	0	no	ac	be	?	?	n.d.	?
157	3 ×	0	no	ad	bc	?	?	n.d.	?
33	3 ×	0	no	ad	be	?	?	n.d.	?
43	3 ×	0	no	ac	be	?	?	n.d.	?
208	3 ×	0	no	ac	be	?	?	n.d.	?

The first two rows show the 2 × seed parent and the 3 × pollen parent, respectively. Only the paternal alleles of the SSR genotypes are shown, since apomixis is inherited from the pollen donor. PAR “+” means that parthenogenesis is confirmed by Differential Interference Contrast (DIC) microscopy. AUT “+” means that autonomous endosperm development is confirmed by DIC microscopy. A question mark (?) means inconclusive; n.q. means not quantified; n.d. means non-determined; FCSS means flow cytometric seed screen. The gray cell fill color shows a positive apomixis component.

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
