# Peer review of "Genetic Dissection of Apomixis in Dandelions Identifies a Dominant Parthenogenesis Locus and Highlights the Complexity of Autonomous Endosperm Formation"

_genes, 2020, doi:10.3390/genes11090961_

Round 1
Reviewer 1 Report
Van Dijk and colleagues have made seminal contributions to understanding the genetic architecture of apomixis, using Taraxacum as a relatively tractable model. Most recently, they succeeded in cloning the dominant locus controlling diplospory/apomeiosis, called DIP, which turned out to encode a VPS13 variant (as reported in a patent application; unfortunately, there is no traditional paper on this yet). VPS13 is a large, deeply conserved protein that has been implicated in vacuolar biogenesis, membrane remodeling and lipid transport; how it may inhibit meiosis remains to be elucidated
Here, the authors describe in detail the segregation of developmental components of apomixis in the progeny of a cross between a diploid, sexual Taraxacum strain with a triploid, apomictic strain used as pollen donor (the same genotypes as in previous work). Likely due to aneuploid pollen, ~98% of the seeds produced in this cross abort; among the 192 viable progeny, 94 triploid plants are recovered and form the basis of the study (the DIP allele promoting apomeiosis is only transmitted in diploid pollen, such that only triplod progeny is informative). ~24% of the triploid F1 showed fertilization-independent seed set with high penetrance, a frequency that is in consistent with a genetic model assuming combined action of three dominant loci (but less so with a two-loci model).
Using a closely linked marker, the authors then identify the 62 triploid plants carrying the DIP allele promoting apomeiosis; investigation of the remaining two developmental components, parthenogenesis and autonomous endosperm development, is limited to this group: the diplosporous gametophytes are genetically uniform, such that the seed phenotypes observed in these plants are not affected by segregation and easier to interpret. Parthenogenesis segregates in a ratio suggesting a single dominant PAR locus, and a linked molecular maker is identified in a survey of 11 co-dominant SSRs – this marks a new and significant result. PAR appears to be unlinked to DIP, and the dominant allele is able to induce embryo development in unfertilized heads of diploid, non-apomictic progeny of the original cross, suggesting an independent mode of action. Autonomous endosperm development is assessed by microscopy as well as ploidy measurements (using flow cytometry) of developing seeds; unexpectedly, the vast majority of plants carrying the DIP allele promoting apomeiosis also show signs of autonomous endosperm. This result implies that apomixis is not determined by three dominant loci and that the genetic basis of autonomous endosperm remains to be determined.
Apomixis is a poorly understood but fascinating from a biological standpoint and of potentially great importance for future agriculture. The study leverages the techniques of classical genetics, including some sophisticated tools (for example, the diploid partner of the original cross carries a cytoplasmic male sterility system, since triploid pollen donors often trigger a breakdown of self-incompatibility) to address this phenomenon. Discovery of a linked molecular marker opens an avenue toward molecular cloning the PAR gene using strategies previously employed in the case of DIP. Although perhaps incremental, the work it was done with great care, and the resulting data is interpreted with authority as well as caution (for example, negative results are treated as missing data). Overall, I found the paper informative, well-conceived, and well-organized. It could appeal to a broad audience.
Minor comments:
Line 246, 315, 359, 360: “F1S“ – should this be “F1”?
Discussion, line 430-435: Wouldn’t these two scenarios (linkage of a dominant automomous endosperm locus with DIP; homozygous autonomous endosperm locus) imply that apomixis should segregate according to a two-locus model?
Figures 5 & 6: The DIC images are rather flat, it would be nice if the contrast could be improved.
Author Response
Reviewer 1
Line 246, 315, 359, 360: "F1S "– should this be "F1"?
Thank you for pointing this out. The corrections have been made accordingly.
Discussion, line 430-435: Wouldn't these two scenarios (linkage of a dominant autonomous endosperm locus with DIP; homozygous autonomous endosperm locus) imply that apomixis should segregate according to a two-locus model?
Thank you for bringing this point up and your hypothesis is correct. However, we observed that the segregation of apomixis as a whole fits a three-locus model and deviates significantly from a two-locus model. If autonomous endosperm formation is somehow fixed in this population as proposed in this paragraph, then existence of a third unidentified dominant locus needs to be postulated to fit our observations of the segregation of apomixis as a whole. To help clarify this understandably confusing point, we have inserted the following sentence:
"If autonomous endosperm formation is fixed in this population, then a developmental function needs to be assigned to the third segregating dominant apomixis locus, yet to be identified." (yellow marked lines 457-459).
To help clarify our results, we have also added a graphical abstract.
Figures 5 & 6: The DIC images are rather flat, it would be nice if the contrast could be improved.
Thank you for this suggestion. We have increased the contrast, which indeed makes the images clearer.
Reviewer 2 Report
The study attempts to discern the allelic basis of the inheritance of apomixis in Taraxacum officinale following a controlled cross and phenotypic analysis of the progeny, coupled to a limited amount of molecular mapping. The population used was quite small for this type of study, particularly given the triploid nature of the apomictic parent and the subsequent need to select progeny based not only on phenotype but also on ploidy. Two molecular markers are described; one in linkage with the DIP locus and another in linkage with the PAR locus. The DIP marker is apparently already reported separately and the new PAR marker is discussed only in terms of its name and approximate linkage distance. No marker sequences were provided making this a hard study to repeat. The observation that the parthenogenesis is controlled by a dominant determinant at a discrete locus is valuable information although this observation is already inferred elsewhere in the literature. The main unique finding, therefore, appears to be that parthenogenesis and autonomous endosperm formation are separately controlled in this system. This is a useful observation, but one the authors were only able to provide a limited amount of information on because of the nature of the data they had to analyze.
Author Response
Reviewer 2
The population used was quite small for this type of study
Thank you for this comment: we fully agree that the population was rather small and would have preferred to use a larger population with more statistical power. However, the production of the population used was a major effort because, as described in the text, there is extremely low pollen fertility from the triploid A68 line. Additionally, to generate 62 crosses required that the many inflorescences had to be hand cross-pollinated within the short-time frame of the synchronous flowering of these specific parents. That said, the population size is around 10-fold higher that what had previously been investigated in the literature for similar crosses, which we believe is sufficiently large for the present analysis and to support the conclusions.
No marker sequences were provided making this a hard study to repeat.
Thank you for bringing this to our attention. While all of the SSR markers used in this study have been previously published as indicated in the materials and methods section, in order to enable independent validation of our results, we have inserted a table with the primer sequences as given by the original papers in which the SSR marker development was described (yellow marked caption Table 1: lines 211-213).
The observation that the parthenogenesis is controlled by a dominant determinant at a discrete locus is valuable information although this observation is already inferred elsewhere in the literature.
Thank you for this observation, and you are correct in that a single dominant determinant is one of hypotheses for control of parthenogenesis that has been proposed in the literature. However, these hypothesis were based on the observation of a few phenotypic recombinants, and, as the reviewer warns in their first comment, these studies are hampered by an even smaller population size than used here. A similar hypothesis of a single dominant locus has been postulated in the literature for autonomous endosperm, which we show here is inconsistent with our observed segregation pattern, highlighting how easy it would have been to reach the incorrect conclusion relying solely on the previous studies. Additionally, our finding of the linked PAR marker proves that PAR is controlled by a dominant allele and that the DIP and PAR loci are not linked. Taken together, we propose that this is a highly significant result, which is helpful for the ongoing efforts to clone the parthenogenesis gene.
The main unique finding, therefore, appears to be that parthenogenesis and autonomous endosperm formation are separately controlled in this system. This is a useful observation, but one the authors were only able to provide a limited amount of information on because of the nature of the data they had to analyze.
Thank you for this opinion, and you are correct that one of our main novel findings is that parthenogenesis and autonomous endosperm are controlled be separate loci. As the reviewer suggests, the control of autonomous endosperm in Taraxacum is observed to be different from what has been found in Hieracium and Erigeron, the other two autonomous apomictic genera in the Asteraceae studied so far. However, our observations also strengthening the lines of evidence supporting a single domain locus controlling parthenogenesis (another unique finding mentioned in the comment above), which will enable the cloning of the PAR gene, a major breakthrough in Apomixis. Lastly, that autonomous endosperm formation is not controlled by a single dominant locus is another unique finding that should not be discounted. In the discussion, we proposed several testable hypotheses to try and understand this further which will stimulate further research and result in a better understanding of the genetic basis of autonomous endosperm.
So, while we fully agree with the reviewer on the challenges that the relatively small population used for this analysis provided (as mentioned above), in spite of these challenges, we believe that the results described here provide for a significant breakthrough in the study of Apomixis, and would be well suited for the “Genes” special issue on the “Molecular Basis of Apomixis in Plants”.
Round 2
Reviewer 2 Report
Thank you for your clarifications